# Nursery teachers in preschool institutions facing burnout: Are personality traits attributing to its development?

Radica Tasic[1], Nina Rajovic[2], Vedrana Pavlovic[2], Bosiljka Djikanovic[3], Srdjan Masic[4], Igor Velickovic[1], Danka Mostic[5], Jelena Cumic[5], Petar Milcanovic[2], Valerija Janicijevic[6], Dejana Stanisavljevic[2], Natasa Milic[2,7] *

**1** Medical School, College of Vocational Studies, Belgrade, Serbia, **2** Institute for Medical Statistics and Informatics, Faculty of Medicine University of Belgrade, Belgrade, Serbia, **3** Department for Public Health, Faculty of Medicine University of Belgrade, Belgrade, Serbia, **4** Department for Public Health, Faculty of Medicine University of East Sarajevo, Foca, Bosnia and Herzegovina, **5** Clinical Centre of Serbia, Belgrade, Serbia, **6** Teacher Education Faculty University of Belgrade, Belgrade, Serbia, **7** Division of Nephrology and Hypertension, Department of Internal Medicine, Mayo Clinic, Rochester, MN, United States of America

* milic.natasa@mayo.edu

**Data Availability Statement:** All relevant data are within the manuscript and its Supporting Information files.

## Abstract

### Introduction

The aim of the study was to assess the prevalence of burnout syndrome among nursery teachers in Belgrade's preschool institutions, and to assess the role of personality traits in its development.

### Materials and methods

A cross-sectional study was conducted in all Belgrade's preschool institutions. A stratified cluster sampling method was used to obtain a representative sample. Burnout was measured using the Maslach Burnout Inventory- General Survey (MBI-GS). The MBI-GS adaptation was based on an internationally accepted methodology for translation and cultural adaptation. Psychometric properties of the Serbian version of the MBI-GS were analyzed through the examination of factorial structure and internal consistency. A shortened version of Big Five Plus Two questionnaire was used to measure Personality traits.

### Results

Three hundred two health care professionals were enrolled. The mean age was 38±9.2 years and all were female. Confirmatory factor analysis validated the three-factor structure of the questionnaire (exhaustion, cynicism and professional efficacy). Overall, 251 (83.1%) respondents were found to have moderate burnout. In multiple regression analysis, positive valence and conscientiousness were significantly associated with professional efficacy. Aggressiveness, neuroticism, openness, and age, were significantly associated with exhaustion. Aggressiveness, neuroticism and additional jobs showed association with cynicism.

**Funding:** The author(s) received no specific funding for this work.

**Competing interests:** The authors have declared that no competing interests exist.

## Conclusion

Present study provided the evidence for the appropriate metric properties of the Serbian version of MBI-GS. Most nurses demonstrated moderate burnout level. Personality traits are characteristics that affect presence of burnout syndrome in healthcare professionals working in preschool institutions.

## Introduction

Alongside higher demands on employees over all professions, especially nursing, the negative impact on nurses' mental health is relentlessly increasing and chronic stress concerning professional disorders is growing into significant issue [1–3]. Child care workers are encountering various everyday' responsibilities, from fulfilling the basic needs (dipper changing, feeding, etc.) of many children several times each hour, to implementing learning activities through a developmentally-appropriate curriculum [4]. Working with young children is very stressful and nurses are at potential risk for burnout syndrome from enduring exposure to immense demands [5].

The "burnout syndrome" has been defined as a state of emotional exhaustion, depersonalization, and reduced personal accomplishment. The term "burnout" was first used in 1974 by Freudenberger in his study called "Stuff burnout" [6]. Shortly after, burnout syndrome was defined by Maslach and Jackson, as a three dimensional syndrome characterized by exhaustion, cynicism and professional inefficacy [7]. Since its release, the Maslash Burnout Inventory (MBI) questionnaire became one of the most commonly used instruments for the assessment of the burnout syndrome. It was translated into many languages and validated worldwide [8]. The need for a general scale to measure workplace combustion regardless the type of work prompted the development of the MBI–General Survey (MBI-GS) [9].

It is considered that personality traits can play an important role in the development of burnout syndrome [10]. People have different personality traits, which can reflect both the working activities and interpersonal relations [11]. The Big Five Plus Two inventory represents a psycholexical model of personality traits, based on non-restrictive methodology by Tellegen and Waller and it is an adequate instrument for measurement of various psychological phenomena [12, 13].

To our knowledge, no empirical research exists addressing the prevalence of a burnout syndrome at work among nursery teachers in preschool institutions in Serbia. According to Bright and Calabro, health and safety concerns of child care workers have been "widely neglected" and should be a focus of more research in this field [14]. Timely recognition of the existence of a burnout syndrome among employees in child care service is of a great importance not only for employees but also for children as direct users of services. Since MBI-GS questionnaire had not been validated for the population of Serbia, the aims of this research were to assess the psychometric properties of the instrument and to assess burnout among nursery teachers, as well as the role of personality traits in the onset of burnout syndrome in the population of healthcare workers in Belgrade's preschool institutions.

## Materials and methods

A cross-sectional study was conducted in all Belgrade's preschool institutions: „11. April"Novi Beograd, „Dr Sima Milošević"Zemun, Čukarica, „Čika Jova Zmaj"Voždovac, Savski venac, Vračar, „Dečji dani"Stari grad, Zvezdara, Rakovica, „Boško Buha"Palilula,

„Poletarac"Barajevo, „Lane"Grocka, „Rakila Kotarov Vuka"Lazarevac, „Jelica Obradović"Mladenovac, „Perka Vićentijević"Obrenovac, „Naša radost"Sopot and Surčin. The participants of the study were nursery teachers. We used a stratified cluster sampling method to obtain a representative sample, where each preschool institution presented a stratum from which clusters of children aged from 6 months to 3 years were randomly selected. Nurses whose work performance was related to the selected clusters entered the study. In the last quarter of 2018, there were 17 registered preschool institutions, 794 groups of children aged 6 months to 3 years, and 1588 nursery teachers currently employed in Belgrade preschool institutions (2 nursery teachers per group). The data were obtained from The Secretariat for Child Protection of the City of Belgrade for Preschool institutions. In this research, we planned to cover 20% of clusters of children aged 6 months to 3 years, which made a total of 302 nursery teachers. The sample size estimation was based on the assumption needed to be fulfilled for factor analysis use, set by Tabachnick and Fidell [15], where the minimum number of respondents must be 150, with at least 5 respondents for each item. Criteria for exclusion from the study were: discontinuity in work for more than one year, such as longer study residencies abroad, longer sick leave or multiple job changes over the last 5 years; exposure to a greater psycho-physical trauma independent of the professional environment, disagreement with participation in the research. Self-reported questionnaires including the Maslach Burnout Inventory- General Survey (MBI-GS) and Big Five Plus Two Questionnaire were distributed. Additionally, demographic, educational background and occupational data were collected.

Burnout was measured using the MBI-GS. The questionnaire was translated into Serbian language and adapted with permission from the copyright owner (Sinapsa Edicija d.o.o., Naklada Slap Group, contract nᵒ 05/2017). The original English language version was translated into Serbian language using the standard forward and backward translation procedure recommended by Wild [16]. Differences between the original and back-translated version were resolved through consensus. The Serbian version of MBI-GS was pre-tested among 20 nursery teachers for clarity, comprehension and understanding. The final version of Serbian MBI-GS was made after modifications made in terms of clarification and simplification of wording, based on participants' feedback.

The MBI-GS is a 16-item scale and assesses three separate aspects of burnout: exhaustion (5 items), cynicism (5 items) and professional efficacy (6 items). Responses to each item are scored on a 7-point Likert scale, ranging from "never"(0) to „everyday"(6). High values of exhaustion and cynicism subscales correspond to high level of burnout, while the professional efficacy is interpreted in the opposite direction. Using the recommended cut-off values, results were categorized as low, moderate and high burnout (Exhaustion: ≤2, 2.01–3.19, and ≥3.2; Cynicism: ≤1, 1.01–2.19, and ≥2.2; Professional Efficacy: ≥5, 4.01–4.99, and ≤4).

A shortened version of Big Five Plus Two questionnaire was used to measure Personality traits [12]. A scale is composed of 70 items which assess five basic (neuroticism, extraversion, openness, conscientiousness, aggression), and two additional dimensions (positive and negative valence) [17]. Responses to each question are scored on a 5 step Likert's scale, ranging from "strongly disagree" (1) to "strongly agree" (5).

Participation was voluntary and anonymous. The approval was obtained from the Education and Child Protection Center of the City of Belgrade and from the Ethics Committee of the Medical Faculty of the University of Belgrade.

## Statistical analysis

Numerical data were presented as means or medians with corresponding measures of variability (ranges, standard deviations or interquartile ranges). Categorical data were presented as

absolute numbers with frequencies. Kolmogorov-Smirnov and Shapiro-Vilkov test were used to test normal distribution. The construct validity of the Serbian version of the MBI-GS was tested using confirmatory factor analysis (CFA). Multiple fit indices were conducted in CFA: Comparative-Fit Index (CFI), Tucker-Lewis index (TLI) and the Root Mean Square Error of Approximation (RMSEA). Internal consistency of the Serbian version of MBI-GS was assessed by using Cronbach alpha coefficient. Pearson's correlation coefficients were calculated to explore the relationship between MBI-GS and Big Five Plus Two subscales. According to Evans' classification (18), a Pearson correlation coefficient $r < 0.20$ was considered to represent a very week correlation, 0.20–0.39 week, 0.40–0.59 moderate, 0.60–0.79 strong, and $> 0.80$ very strong correlation. Multiple linear regression analysis was used to determine factors related to burnout. MBI-GS subscales were used as dependent variables in separate regression models. Independent variables were the following: sex, age, marital status, level of education, length of service, managerial positions, socioeconomic status, and personality type. For the implementation of multiple linear regressions, model assumptions were taken into account. All tests were two-tailed. $P < 0.05$ was considered statistically significant. Statistical analysis was done using Amos 21 (IBM SPSS Inc., Chicago, IL, 2012) and IBM SPSS Statistics 25 software.

## Results

The Serbian version of MBI-GS and Big Five Plus Two questionnaires were completed by 302 nurses employed in Belgrade preschool institutions. The mean age of nurses was 38±9.2 (range 16–60 years) and they were all female. Average length of service was 11 years (IQR 6–18) and most of them had secondary school qualifications (79.7%). About half of participants reported average socio-economic status (52.5%), and 93.6% of them worked on non-executive positions (Table 1).

The confirmatory analyses showed that the three dimensional model fit increased when Item 13 was omitted, providing a sound fit to the observed data. Item 13 had an extremely low factor loading on the cynicism dimension, and was therefore removed. The values for fit indices TLI (0.849) and CFI (0.875) were close to their cutoff criteria (0.90). The RMSEA value of 0.104 (0.093–0.115) was over recommended 0,8, as is often observed in large samples. Standardized factor loadings were statistically significant and ranged from 0.30 to 0.85 (Fig 1). The

**Table 1. Characteristics of the study sample.**

| Variable | n (%) |
|---|---|
| Female | 302 (100) |
| Age, yrs* | 38.5±9.2 (16–60) |
| Length of service, yrs** | 11 (6–18) |
| Level of professional qualifications: | |
| Secondary School Qualifications | 239 (79.7) |
| Two-year Post-secondary School Qualifications | 22 (7.3) |
| University Qualifications | 39 (13.0) |
| Executive position | 19 (6.4) |
| Non-executive position | 280 (93.6) |
| Socio-economic status | |
| Low | 23 (7.8) |
| Average | 155 (52.5) |
| High | 117 (39.7) |

* Data are presented as mean±SD (min-max)

**Data are presented as median (25–75 percentiles)

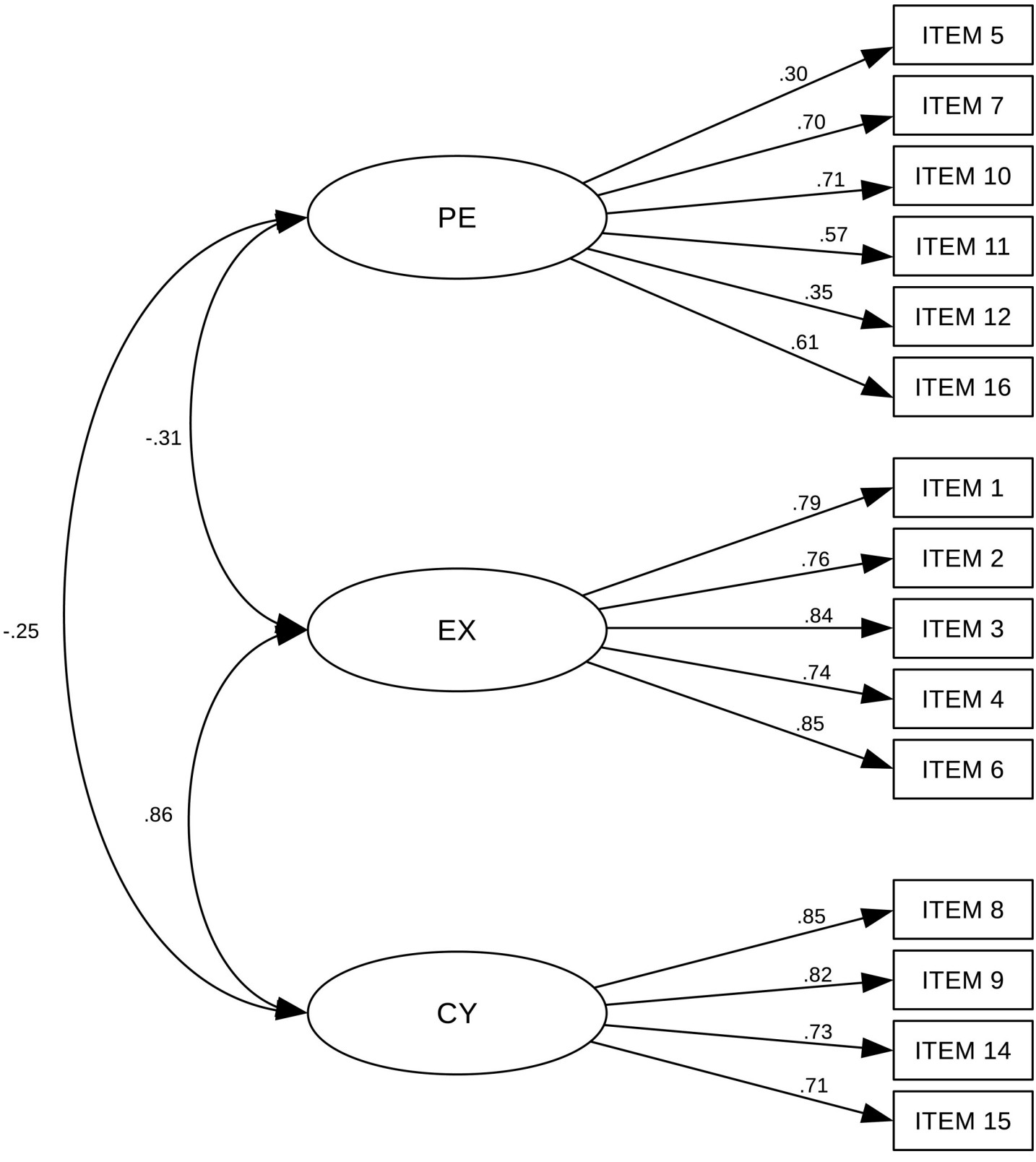

**Fig 1. Standardized factor loadings for MBI-GS questionnaire in Serbian language.**

**Table 2. Internal consistency of MBI-GS and Big Five Plus Two questionnaire.**

| Questionnaire | Cronbach's Alpha | Internal consistency |
|---|---|---|
| **MBI-GS** | | |
| Professional efficacy | 0.640 | Questionable |
| Exhaustion | 0.890 | Good |
| Cynicism | 0.726 | Acceptable |
| Total | 0.774 | Acceptable |
| **Big Five Plus Two** | | |
| Aggressiveness | 0.759 | Acceptable |
| Extraversion | 0.791 | Acceptable |
| Neuroticism | 0.764 | Acceptable |
| Negative Valence | 0.808 | Good |
| Openness | 0.661 | Questionable |
| Positive Valence | 0.767 | Acceptable |
| Conscientiousness | 0.715 | Acceptable |
| Total | 0.759 | Acceptable |

correlation between Exhaustion and Cynicism subscales was strong positive, while there was week negative relationship between Exhaustion and Professional efficacy and no relationship between Professional efficacy and Cynicism. Cronbach's alpha coefficient of the overall 16-item MBI-GS was 0.774, which indicates that the scale has acceptable reliability (Table 2). The Cronbach's alpha coefficients of Professional efficacy, Exhaustion and Cynicism subscales were 0.640, 0.890 and 0.726, respectively. Analysis of internal consistency of The Big Five Plus Two questionnaire showed that in total Cronbach's alpha was 0.759, indicating scale reliability. The alpha coefficients of The Big Five Plus Two subscales ranged from 0.661 to 0.808.

In this study, the average subscale score for exhaustion was 1.5 ± 1.3. Moderate to high level of exhaustion was observed in 27.6% of nursery teachers (Table 3). The average subscale score for professional efficacy was 5.3 ± 0.7, and the scores were 78.8%, 16.1% and 5.1% in the ranges of low, moderate and high burnout, respectively. The average subscale score for Cynicism was 1.6±1.1. According to this subscale results, most nursery teachers had moderate and high level of burnout: 56.6% and 21.0%, respectively. Overall, 251 (83.1%) nurses were found to have moderate burnout. Four nurses fell into the category of high burnout based on high exhaustion, high cynicism and low professional efficacy scores, and 47 (15.6%) nurses scores showed low burnout (low exhaustion, low cynicism, high professional efficacy).

High mean subscale scores of Big Five Plus Two questionnaire were found for Extraversion (4.2±0.5), Conscientiousness (4.4±0.5), and for Openness (4.0±0.5), while low average subscale scores were found for Aggressiveness (1.8±0.6), Neuroticism (1.7±0.6) and Negative Valence

**Table 3. Burnout among nursery teachers in Belgrade's preschool institutions.**

| Domain | Low | Moderate | High |
|---|---|---|---|
| Exhaustion | 215 (72.4%) | 46 (15.5%) | 36 (12.1%) |
| Cynicism | 66 (22.4%) | 167 (56.6%) | 62 (21.0%) |
| Professional efficacy | 230 (78.8%) | 47 (16.1%) | 15 (5.1%) |
| **Overall burnout**[#] | 47 (15.6%) | 251 (83.1%) | 4 (1.3%) |

[#]**High burnout**: High Exhaustion, High Cynicism, Low Professional efficacy; **Low burnout**: Low Exhaustion, Low Cynicism, High Professional efficacy

**Table 4. Correlation coefficients (r) between burnout, personality traits and demographic characteristics.**

| | Mean ±SD | 1 | 2 | 3 | 4 | 5 | 6 | 7 | 8 | 9 | 10 | 11 | 12 | 13 | 14 |
|---|---|---|---|---|---|---|---|---|---|---|---|---|---|---|---|
| **Burnout dimensions** | | | | | | | | | | | | | | | |
| 1. Professional Efficacy | 5.3±0.7 | | | | | | | | | | | | | | |
| 2. Exhaustion | 1.5±1.3 | -0.274* | | | | | | | | | | | | | |
| 3. Cynicism | 1.6±1.1 | -0.094 | 0.706* | | | | | | | | | | | | |
| **Personality traits** | | | | | | | | | | | | | | | |
| 4. Aggressiveness | 1.8±0.6 | -0.077 | 0.334* | 0.291* | | | | | | | | | | | |
| 5. Extraversion | 4.2±0.5 | 0.274* | -0.278* | -0.196* | -0.275* | | | | | | | | | | |
| 6. Neuroticism | 1.7±0.6 | -0.208* | 0.418* | 0.287* | 0.376* | -0.442* | | | | | | | | | |
| 7. Negative Valence | 1.2±0.3 | -0.096 | 0.203* | 0.191* | 0.488* | 0.282* | 0.342* | | | | | | | | |
| 8. Openness | 4.0±0.5 | 0.274* | -0.265* | -0.170* | -0.176* | 0.525* | 0.236* | -0.234* | | | | | | | |
| 9. Positive Valence | 3.3±0.7 | 0.258* | -0.172* | -0.084 | 0.105 | 0.325* | -0.147* | 0.099 | 0.310* | | | | | | |
| 10. Conscientiousness | 4.4±0.5 | 0.275* | -0.303* | -0.250* | 0.260* | 0.465* | -0.344* | -0.354* | 0.340* | 0.185* | | | | | |
| **Demographic characteristics** | | | | | | | | | | | | | | | |
| 11. Age | | -0.062 | 0.220* | 0.180* | 0.095 | -0.080 | 0.217* | 0.122 | -0.142* | -0.017 | -0.142* | | | | |
| 12. Length of service | | -0.066 | 0.212* | 0.192* | 0.107 | -0.121* | 0.207* | 0.080 | -0.141* | -0.060 | -0.185* | 0.758* | | | |
| 13. Executive/ Non-executive position | | 0.047 | -0.008 | -0.024 | 0.089 | -0.086 | 0.050 | 0.089 | 0.035 | 0.035 | -0.025 | 0.081 | 0.127* | | |
| 14. Number of children | | -0.067 | 0.094 | 0.045 | 0.002 | 0.029 | 0.079 | 0.051 | -0.139 | 0.043 | 0.020 | 0.550* | 0.319* | 0.022 | |
| 15. Material state | | 0.087 | -0.087 | -0.040 | -0.002 | 0.148* | 0.290* | -0.023 | 0.156* | 0.082 | 0.159* | -0.224* | -0.165* | -0.039 | -0.146* |

r<0.20, very week; 0.20–0.39, week; 0.40–0.59, moderate; 0.60–0.79 strong; >0.80, very strong correlation [18]

(1.2±0.3). Mean subscale score for Positive Valence (3.3±0.7) was moderate. The correlations between Big Five personality traits and burnout are presented in Table 4. Overall, big five personality traits were strongly correlated with burnout. As is shown in this table, aggressiveness, neuroticism and negative valence were positively correlated with exhaustion and cynicism. The traits of extraversion, openness, positive valence and conscientiousness were positively correlated with professional efficacy, but negatively correlated with exhaustion and cynicism, except for positive valence and cynicism. Age and lenght of service were found to show significantly positive correlation with exhaustion and cynicism.

In multiple regression analysis, positive valence and conscientiousness were significantly associated with professional efficacy. Nursery teachers with higher values of positive valence and conscientiousness had higher professional efficacy scores compared to those with lower results obtained for these two traits. Aggressiveness, neuroticism, openness, and age, in a multiple regression model, were significantly associated with exhaustion. Older nursery teachers with higher values of aggressiveness, neuroticism, and openness had higher exhaustion scores compared to younger nurses and those with lower results obtained for these traits. Aggressiveness, neuroticism and additional jobs showed association with cynicism. Nursery teachers with higher values of aggressiveness and neuroticism, and those with additional jobs had higher cynicism scores compared to nurses without additional jobs and those with lower results obtained for these two traits (Table 5).

## Discussion

Most nurses demonstrated moderate burnout level. The present study examined the prevalence of burnout and its connection with personality traits among nursery teachers, and to our

**Table 5. Multiple linear regression analysis with MBI-GS subscales as dependent variables.**

| | β | t | p |
|---|---|---|---|
| **Professional Efficacy** | | | |
| Extraversion | 0.110 | 1.348 | 0.179 |
| Neuroticism | -0.08 | -1.12 | 0.911 |
| Openness | 0.041 | 0.559 | 0.577 |
| Positive Valence | 0.221 | 3.368 | **0.001** |
| Conscientiousness | 0.188 | 2.596 | **0.010** |
| Level of professional qualifications | 0.117 | 1.881 | 0.061 |
| **Exhaustion** | | | |
| Aggressiveness | 0.220 | 3.052 | **0.003** |
| Extraversion | 0.124 | 1.551 | 0.123 |
| Neuroticism | 0.365 | 4.907 | **0.000** |
| Negative Valence | -0.114 | -1.581 | 0.116 |
| Openness | -0.193 | -2.544 | **0.012** |
| Positive Valence | -0.084 | -1.234 | 0.219 |
| Conscientiousness | -0.048 | -0.681 | 0.497 |
| Age, yrs | 0.212 | 2.156 | **0.032** |
| Length of service, yrs | -0.049 | -0.500 | 0.618 |
| Additional jobs | 0.121 | 1.955 | 0.052 |
| **Cynicism** | | | |
| Aggressiveness | 0.222 | 2.938 | **0.004** |
| Extraversion | 0.037 | 0.440 | 0.660 |
| Neuroticism | 0.196 | 2.496 | **0.013** |
| Negative Valence | -0.074 | -0.989 | 0.324 |
| Openness | -0.130 | -1.659 | 0.099 |
| Conscientiousness | -0.072 | -0.968 | 0.334 |
| Age, yrs | 0.150 | 1.452 | 0.148 |
| Length of service, yrs | -0.021 | -0.201 | 0.841 |
| Additional jobs | 0.191 | 2.901 | **0.004** |

knowledge, it is the first cross-sectional questionnaire based study conducted in Serbia on this population.

This study's results show that average age of our participants is 38.5 years, which represents significantly lower average age compared to the similar study conducted in Andalusia, Spain (mean age: 44.58 years), and significantly higher average age compared to study performed in Marmara region of Tukey, where mean age of nurses was 28.95 years [19, 20]. Some of the studies indicated that age did not have any effect on burnout level [21], while study of Sekol and Kim presented opposite results [22]. In addition to this, study published by Berger at al. [23], showed that burnout was associated with the nurses' age and that younger nurses were more prone to burnout. Their results do not correspond to results obtained in our study, where positive correlation of nurses' age with two domains of burnout syndrome, Exhaustion and Cynicism, was found. It is possible that there is an influence of age on burnout level among nurses because of advantages of the acquisition of experience and upgraded workplace problem solving [20].

Neither in our nor in similar study conducted in Turkey [20] other demographic variables, such as number of children and level of nurses' education predicted significant burnout levels (p>0.001). Average duration of working as nursery teacher was 11 years and it predicted significant burnout levels, opposite to study results obtained in Marmara region of Turkey [20].

As presented in a results section, very few nurses met criteria for high burnout and low burnout. Using the cut-off values obtained from the 4th edition of the Maslach handbook, our results showed that majority of nursery teachers belonged to moderate burnout group (83.1%). Similar results were obtained in a systematic review and meta-analysis of 34 studies concerning burnout in pediatric nurses, carried out by Hernandez et al. [24], showing that a significant number of pediatric nurses were found to have moderate to high levels of burnout. These nurses, therefore, were either experiencing burnout or at high risk of suffering it in the future. Urgency for further study of workplace hazards which are leading to burnout syndrome are supported by previously mentioned studies [25]. They also emphasis the priority of developing and improving interventions and therapies to prevent the above symptoms, in this manner helping nurses to cope with situations and circumstances that may lead to burnout syndrome [24].

Even though burnout mainly refers to both situational and individual factors, including personality, scarcely any studies have recognized the connection between personality traits and burnout in nurses. In a cross-sectional study conducted in 2016 in Australia, 140 eligible neonatal nurses provided the data. Self-reported questionnaires were used to measure burnout and five-factor model of personality traits, where Extraversion and Neuroticism showed relation to burnout [26]. Neuroticism and Extraversion showed correlation with all three domains of burnout syndrome in our study: Professional Efficacy, Cynicism and Exhaustion (p<0.001). Association of high Neuroticism with burnout syndrome was previously reported in study of Vlerick in 2001 in Belgian nurses, as well as in US nurses in study of Zellars et al. [27, 28]. Study performed in Almeria, Spain analyzed the relationship between certain personality traits and the presence of burnout in nursing personnel. Their study' results showed that burnout in this group of healthcare workers was associated negatively with Extraversion, Conscientiousness and Openness, but had a positive relationship with Neuroticism [29]. Although numerous studies examining burnout and personality traits found correlations between the two, the relevance of statistical significance for its practical consideration should be further addressed. Qualitative research determining the relation of various factors contributing to the development of burnout could provide better insight into understanding the influence of personality on burnout.

Health professionals are common occupational groups constantly explored in the field of burnout syndrome. Notion of nursing as occupation prone to burnout support the high expectations while working with young children, emotionally exhausting demands, immense workload and stressful conditions for employees [30]. In addition, contact with parents and guardians, who are getting more demanding, should not be neglected as an important trigger of professional exhaustion in the nursing occupation. Very little, nonetheless, is known about the development of burnout in nursing teachers working in preschool institutions, as well as the impact of personality traits on its development. Alarcon et al in 2009 reported that 'even when organizations use burnout interventions that focus on changing work environment, by reducing or eliminating job stressors, some individuals may still experience high levels of burnout as a result of their personalities' [31]. Personality traits are characteristics that effect behavior and approach of healthcare professionals to daily circumstances. It is an enduring challenge to recognize how personality of the individual can affect the development of burnout, thus the relationship between the sociodemographic, work and personality factors in nursing personnel must be known in order to understand the presence of this ever more prevalent phenomenon [32]. The results of this and similar research should encourage decision-makers to devote efforts for solving this complex problem by a deeper understanding of factors contributing to the presence of burnout syndrome among health care professionals working with young children.

## Conclusion

Present study provided the evidence for the appropriate metric properties of the Serbian version of MBI-GS. Most nurses demonstrated moderate burnout level. Personality traits are characteristics that affect presence of burnout syndrome in healthcare professionals working in preschool institutions.

## Supporting information

**S1 File. Minimal data set.**
(XLS)

## Acknowledgments

The authors are thankful to Education and Child Protection Center of the City of Belgrade and Medical Faculty of the University of Belgrade for giving permission to conduct this study and the administrative support. We are most grateful to the Sinapsa Edicija d.o.o. for assistance in obtaining permission to use MBI-GS questionnaire.

## Author Contributions

**Conceptualization:** Radica Tasic, Nina Rajovic, Vedrana Pavlovic, Bosiljka Djikanovic, Srdjan Masic, Danka Mostic, Jelena Cumic, Petar Milcanovic, Valerija Janicijevic, Dejana Stanisavljevic, Natasa Milic.

**Data curation:** Radica Tasic, Nina Rajovic, Vedrana Pavlovic, Igor Velickovic, Danka Mostic, Jelena Cumic, Natasa Milic.

**Formal analysis:** Radica Tasic, Nina Rajovic, Vedrana Pavlovic, Srdjan Masic, Igor Velickovic, Petar Milcanovic, Valerija Janicijevic, Dejana Stanisavljevic, Natasa Milic.

**Funding acquisition:** Radica Tasic, Dejana Stanisavljevic, Natasa Milic.

**Investigation:** Radica Tasic, Nina Rajovic, Vedrana Pavlovic, Bosiljka Djikanovic, Srdjan Masic, Igor Velickovic, Danka Mostic, Jelena Cumic, Petar Milcanovic, Valerija Janicijevic, Dejana Stanisavljevic, Natasa Milic.

**Methodology:** Radica Tasic, Nina Rajovic, Vedrana Pavlovic, Bosiljka Djikanovic, Srdjan Masic, Igor Velickovic, Danka Mostic, Jelena Cumic, Petar Milcanovic, Valerija Janicijevic, Dejana Stanisavljevic, Natasa Milic.

**Project administration:** Natasa Milic.

**Resources:** Dejana Stanisavljevic, Natasa Milic.

**Software:** Dejana Stanisavljevic, Natasa Milic.

**Supervision:** Bosiljka Djikanovic, Valerija Janicijevic, Dejana Stanisavljevic, Natasa Milic.

**Validation:** Nina Rajovic, Vedrana Pavlovic, Bosiljka Djikanovic, Igor Velickovic, Danka Mostic, Jelena Cumic, Petar Milcanovic, Valerija Janicijevic, Dejana Stanisavljevic, Natasa Milic.

**Visualization:** Radica Tasic, Nina Rajovic, Vedrana Pavlovic, Bosiljka Djikanovic, Srdjan Masic, Igor Velickovic, Danka Mostic, Petar Milcanovic, Dejana Stanisavljevic, Natasa Milic.

**Writing – original draft:** Radica Tasic, Nina Rajovic, Vedrana Pavlovic, Bosiljka Djikanovic, Srdjan Masic, Igor Velickovic, Danka Mostic, Jelena Cumic, Petar Milcanovic, Valerija Janicijevic, Dejana Stanisavljevic, Natasa Milic.

**Writing – review & editing:** Radica Tasic, Nina Rajovic, Vedrana Pavlovic, Bosiljka Djikanovic, Srdjan Masic, Igor Velickovic, Danka Mostic, Jelena Cumic, Petar Milcanovic, Valerija Janicijevic, Dejana Stanisavljevic, Natasa Milic.

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
