## [Decision Letter · Decision Letter 0]

22 Oct 2020

PONE-D-20-20845

Nursery teachers in preschool institutions facing burnout: Are personality traits attributing to its development?

PLOS ONE

Dear Dr. Milic,

Thank you for submitting your manuscript to PLOS ONE. After careful consideration, we feel that it has merit but does not fully meet PLOS ONE’s publication criteria as it currently stands. Therefore, we invite you to submit a revised version of the manuscript that addresses the points raised during the review process.

We look forward to receiving your revised manuscript.

Kind regards,

Stefan Hoefer

Academic Editor

PLOS ONE

Journal Requirements:

Reviewers' comments:

Reviewer's Responses to Questions

**Comments to the Author**

1. Is the manuscript technically sound, and do the data support the conclusions?

Reviewer #1: Yes

Reviewer #2: Partly

2. Has the statistical analysis been performed appropriately and rigorously? 

Reviewer #1: Yes

Reviewer #2: Yes

3. Have the authors made all data underlying the findings in their manuscript fully available?

Reviewer #1: Yes

Reviewer #2: Yes

4. Is the manuscript presented in an intelligible fashion and written in standard English?

Reviewer #1: Yes

Reviewer #2: Yes

5. Review Comments to the Author

Reviewer #1: Reviewer comments

Nursery teachers in preschool institutions facing burnout: Are personality traits attributing to its development?

This publication is written correctly and clearly. As a pediatrician and the head of the children department I know that working with young children is very stressful the new element in this work is more difficult contact with parents and relatives Therefore, I consider it important to take up the subject of burnout.

The scheme of article is logical and the discussion section is written widely with good conclusions and suggestions. So far, only a few reports have been reported in the literature. An additional advantage of the work is the large, well-described group of participants .This makes the conclusions presented are plausible . Reliable statistical methods were used.

I recommend the article for publication

I only have a few minor comments:

33-35

My proposition is to put this information at the end of publication

287 You a re right that the notion of nursing as occupation prone to burnout support the high expectations while working with young children, emotionally exhausting demands, immense workload and stressful conditions for employees However, she proposes to emphasize the role of contact with parents and guardians of children, as they are more and more aggressive and this has an impact on professional exhaustion.

My last remark I suggests the new publication , based on similar tests, two years after the COVID epidemic began.

Reviewer #2: This is exciting work showing great promise regarding the attention to specific issues within the field of supporting those working with very young children. The analysis of the data is clear and well presented. My only suggestion for a minor modification is to clarify the distinction between "statistical significance" and "practical significance". While the statistical significance was clear throughout, the practical significance was not. For example, the strength of the negative relation between professional efficacy and exhaustion was rather low compared to the much stronger relation between exhaustion and cynicism. A question that can be raised is does one cause the other or do the two simply coexist? This raises the need perhaps, for qualitative research to further investigate the relation among the various factors examined that tend to show a relation. Near the end of the paper was a hint of addressing this by suggesting a more personalized approach, but I am not sure. Perhaps this could be clarified buy the consideration of the distinction of practical significance between the stronger associations identified in this study.

6. PLOS authors have the option to publish the peer review history of their article (what does this mean?). If published, this will include your full peer review and any attached files.

Reviewer #1: No

Reviewer #2: No

---

## [Author Response · Author response to Decision Letter 0]

30 Oct 2020

Answers to Reviewers

Reviewer #1: 

Nursery teachers in preschool institutions facing burnout: Are personality traits attributing to its development?

This publication is written correctly and clearly. As a pediatrician and the head of the children department I know that working with young children is very stressful the new element in this work is more difficult contact with parents and relatives Therefore, I consider it important to take up the subject of burnout.

The scheme of article is logical and the discussion section is written widely with good conclusions and suggestions. So far, only a few reports have been reported in the literature. An additional advantage of the work is the large, well-described group of participants .This makes the conclusions presented are plausible. Reliable statistical methods were used.

I recommend the article for publication

I only have a few minor comments:

Q1: 33-35

My proposition is to put this information at the end of publication

A1: Thank you for this remark. We removed this info from abstract lines 33-35, and acknowledged the support of Education and Child Protection Center of the City of Belgrade and the Medical Faculty of the University of Belgrade at the end of the paper.

Q2: You are right that the notion of nursing as occupation prone to burnout support the high expectations while working with young children, emotionally exhausting demands, immense workload and stressful conditions for employees. However, she proposes to emphasize the role of contact with parents and guardians of children, as they are more and more aggressive and this has an impact on professional exhaustion.

A2: Thank you for this remark. We added following comment in this paragraph of discussion: 

In addition, contact with parents and guardians, who are getting more demanding, should not be neglected as an important trigger of professional exhaustion in the nursing occupation.

Q3: My last remark I suggests the new publication, based on similar tests, two years after the COVID epidemic began.

A3: Thank you very much for your kind suggestion and support of our work. We discussed this possibility and agreed that it would be very important and interesting to perform similar study after the two years of COVID appearance. Maybe even after one… we agreed to start working on obtaining permissions for that. 

Reviewer #2: 

This is exciting work showing great promise regarding the attention to specific issues within the field of supporting those working with very young children. The analysis of the data is clear and well presented. 

Q1: My only suggestion for a minor modification is to clarify the distinction between "statistical significance" and "practical significance". While the statistical significance was clear throughout, the practical significance was not. For example, the strength of the negative relation between professional efficacy and exhaustion was rather low compared to the much stronger relation between exhaustion and cynicism. A question that can be raised is does one cause the other or do the two simply coexist? This raises the need perhaps, for qualitative research to further investigate the relation among the various factors examined that tend to show a relation. 

A1: We thank to reviewer for this suggestion. We added the strength of correlation in abovementioned results part, as well as in the Table 4. footnote. Classification used for defining the strength of correlation is added in Methods with appropriate reference to it. Need for further qualitative research in this area is also commented in discussion part. 

Q2: Near the end of the paper was a hint of addressing this by suggesting a more personalized approach, but I am not sure. Perhaps this could be clarified buy the consideration of the distinction of practical significance between the stronger associations identified in this study.

A2: This sentence is reorganized to exclude personalized approach.

---

## [Editor Report · Decision Letter 1]

5 Nov 2020

Nursery teachers in preschool institutions facing burnout: are personality traits attributing to its development?

PONE-D-20-20845R1

Dear Dr. Milic,

We’re pleased to inform you that your manuscript has been judged scientifically suitable for publication and will be formally accepted for publication once it meets all outstanding technical requirements.

Kind regards,

Stefan Hoefer

Academic Editor

PLOS ONE
---

## [Editor Report · Acceptance letter]

11 Nov 2020

PONE-D-20-20845R1 

Nursery teachers in preschool institutions facing burnout: are personality traits attributing to its development? 

Dear Dr. Milic:

I'm pleased to inform you that your manuscript has been deemed suitable for publication in PLOS ONE. Congratulations! Your manuscript is now with our production department. 

Kind regards, 

on behalf of

Dr. Stefan Hoefer 

Academic Editor

PLOS ONE